# High Yield of Functional Dopamine-like Neurons Obtained in NeuroForsk 2.0 Medium to Study Acute and Chronic Rotenone Effects on Oxidative Stress, Autophagy, and Apoptosis

**DOI:** 10.3390/ijms242115744

**Published:** 2023-10-30

**Authors:** Diana Alejandra Quintero-Espinosa, Carlos Velez-Pardo, Marlene Jimenez-Del-Rio

**Affiliations:** Neuroscience Research Group, Institute of Medical Research, Faculty of Medicine, University of Antioquia, University Research Headquarters, Calle 62#52-59, Building 1, Laboratory 411/412, Medellin 050010, Colombia; dalejandra.quintero@udea.edu.co (D.A.Q.-E.); calberto.velez@udea.edu.co (C.V.-P.)

**Keywords:** apoptosis, autophagy, autophagosomes, lysosomes, NeuroForsk 2.0, oxidative stress, rotenone

## Abstract

Several efforts to develop new protocols to differentiate in in vitro human mesenchymal stromal cells (hMSCs) into dopamine (DA) neurons have been reported. We have formulated NeuroForsk 2.0 medium containing fibroblast growth factor type beta (FGFb), brain-derived neurotrophic factor (BDNF), melatonin, purmorphamine, and forskolin. We report for the first time that menstrual stromal cells (MenSCs) cultured in NeuroForsk 2.0 medium for 7 days transdifferentiated into DA-like neurons (DALNs) expressing specific DA lineage markers tyrosine hydroxylase-positive cells (TH+) and DA transporter-positive (DAT+) cells and were responsive to DA-induced transient Ca^2+^ influx. To test the usefulness of this medium, DALNs were exposed to rotenone (ROT), a naturally occurring organic neurotoxin used extensively to chemically induce an in vitro model of Parkinson’s disease (PD), which is a movement disorder characterized by the specific loss of DA neurons. We wanted to determine whether ROT induces apoptotic cell death and autophagy pathway under acute or chronic conditions in DALNs. Here, we report that acute ROT exposure induced several molecular changes in DALNS. ROT induced a loss of mitochondrial membrane potential (ΔΨ_m_), high expression of parkin (PRKN), and high colocalization of dynamin-related protein 1 (DRP1) with the mitochondrial translocase of the outer membrane of mitochondria 20 (TOMM20) protein. Acute ROT also induced the appearance of DJ-1Cys^106^-SO_3_, as evidenced by the generation of H_2_O_2_ and oxidative stress (OS) damage. Remarkably, ROT triggered the phosphorylation of leucine-rich repeat kinase 2 (LRRK2) at residue Ser^935^ and phosphorylation of α-Syn at residue Ser^129^, a pathological indicator. ROT induced the accumulation of lipidated microtubule-associated protein 1B-light chain 3 (LC3B), a highly specific marker of autophagosomes. Finally, ROT induced cleaved caspase 3 (CC3), a marker of activated caspase 3 (CASP3) in apoptotic DALNs compared to untreated DANLs. However, the chronic condition was better at inducing the accumulation of lysosomes than the acute condition. Importantly, the inhibitor of the LRRK2 kinase PF-06447475 (PF-475) almost completely blunted ROT-induced apoptosis and reduced ROT-induced accumulation of lysosomes in both acute and chronic conditions in DALNs. Our data suggest that LRRK2 kinase regulated both apoptotic cell death and autophagy in DALNs under OS. Given that defects in mitochondrial complex I activity are commonly observed in PD, ROT works well as a chemical model of PD in both acute and chronic conditions. Therefore, prevention and treatment therapy should be guided to relieve DALNs from mitochondrial damage and OS, two of the most important triggers in the apoptotic cell death of DALNs.

## 1. Introduction

Menstrual stromal cells (MenSCs) are multipotent cells with self-renewal ability and differentiating potential into mesodermal, endodermal, and ectodermal cell lineages [1,2]. Given that MenSCs can be obtained from menstrual blood (MensB) with minimal technical limitations, minor ethical issues, and/or reduced tumorigenic risk, these cells have emerged as a useful source for 2D in vitro modeling [3,4] and a potential source for PD treatment (e.g., [5,6]). We have derived MenSCs from menstrual blood to obtain dopamine-like neurons (DALNs) in NeuroForsk medium in 7 days [7]. Therefore, it is critical to increase the yield of this cell lineage for in vitro experiments. Nonetheless, finding the optimal combination of molecules for obtaining maximum yield of DA neurons might be a difficult task. Several efforts to develop new protocols to differentiate human pluripotent stem cells (hPSCs) or human mesenchymal stem cells (hMSCs) into DA neurons have recently been reviewed. In particular, a comprehensive analysis of protocols has shown that a total of 27 different small molecules induce DA neurons, including cellular factors and synthetic compounds [8]. Remarkably, the protocols to obtain DA neurons derived from hPSCs and from hMSCs have yielded moderated percentages of TH+ cells [9,10,11,12,13] depending on the biological source (e.g., human Wharton’s jelly MSCs, dental pulp). Inspired by these reports, we modified the NeuroForsk medium into NeuroForsk 2.0 medium by adding fibroblast growth factor type beta (FGFb), brain-derived neurotrophic factor (BDNF), melatonin, and purmorphamine. We report for the first time that MenSCs cultured in NeuroForsk 2.0 medium for 7 days transdifferentiated into DALNs (tyrosine hydroxylase-positive (TH+) and dopamine transporter-positive (DAT+) cells) assessed by flow cytometry analysis. Compared to regular culture medium (RCm), NeuroForsk 2.0 medium yielded a higher percentage of TH+/DAT+ DALNs. Furthermore, DALNs responded to DA stimuli displaying two maximal fluorescence changes (ΔF/F) of Ca^2+^ influx at 10 s and after 80 s of DA addition in NeuroForsk 2.0. Cells did not respond to DA in RCm.

Rotenone (ROT, benzopyrano(3,4-b) furo(2,3-h) (1) benzopyran-6 (6aH)-one, 1,2,12,12a-tetrahydro-2-a-isopropenyl-8,9-dimethoxy) is a naturally occurring organic heteropentacyclic compound and is found in the roots of the *Derris* [14,15] and *Lonchocarpus* [16] plant species. Since ROT offers a broad spectrum of insecticidal, acaricidal, and pesticidal properties, it has attained worldwide fame (e.g., nominated the molecule of the month January 2004; http://www.chm.bris.ac.uk/motm/rotenone/, accessed on 24 October 2023). Importantly, ROT has been associated with the development of Parkinson’s disease (PD, [17,18]). This neurological disorder is a chronic, progressive, and disabling neurodegenerative disorder characterized by motor impairment, autonomic dysfunction, and non-motor features such as psychological and cognitive changes. PD occurs as a consequence of the specific loss of at least 60% of pigmented DA neurons in the substantia nigra pars compacta (SNpc) [19,20]. Neuropathologically, PD is characterized by the formation of the Lewy body, which is composed of eosinophilic cytoplasmic inclusions of aggregated α-synuclein (α-Syn) protein found in DA neurons as well as in other neurons [21,22]. Importantly, ROT induces specific degeneration of DA neurons and aggregation of α−Syn in vitro and in vivo [23,24,25], contributing to cell death [26]. However, the mechanism by which ROT causes vulnerability and DA neuronal cell death has not been yet definitely determined. Several reasons may account for this odd outcome, such as differences in cellular sources (primary versus cell lines), time of exposition (acute versus chronic), dose (low versus high), mode of cell death (necrosis, apoptosis, and autophagy, among others), and/or type of model (in vitro versus in vivo). Furthermore, the effect of ROT or its by-product hydrogen peroxide (H_2_O_2_) on autophagy might depend on the cellular system analyzed. Indeed, it can enhance autophagy and induce autophagic cell death via reactive oxygen species (ROS)-mediated mechanisms in HeLa cells [27] and in human neuroblastoma cell line SH-SY5Y [28]. In contrast, it fails to significantly increase levels of ROS or autophagy in primary mouse astrocytes [27]. ROT-induced ROS can inhibit autophagic flux in differentiated SH-SY5Y cells and PC-12 cells [29,30]. Also, ROT induced oxidative stress (OS), mitochondrial damage, and apoptotic cell death independently of the accumulation of lysosomes and autophagolysosomes in HEK-293 cells [31]. Therefore, the mechanism by which ROT induces cell death or autophagy cell death on DA neurons remain elusive. Answering whether ROT induces primarily alterations in autophagy or apoptosis in DA neurons in PD is critical for therapeutic purposes [32,33]. To solve the divergent observations on the effect of ROT in different cellular systems, we used MenSC-derived DALNs in NeuroForsk 2.0 medium.

Several reports have concluded that ROT acts as a strong inhibitor of complex I of the mitochondrial respiratory chain via inhibition of electron transfer from the iron–sulfur centers in complex I to ubiquinone, leading to a blockade of the I_Q_ site [34,35]. Indeed, overreduction of complex I causes electrons to leak and produce ROS such as the superoxide anion radical (O_2_^−^). The O_2_^−^ radical can dismutate into non-radical reactive H_2_O_2_ [36,37]. In turn, this last compound, via signaling mechanisms [38,39], induces regulated cell death—apoptosis. We and others have shown that ROT-induced concentration- and time-dependent increases in nuclei fragmentation, mitochondrial dysfunction, ROS production, and apoptosis occur in the human neuroblastoma cell line SH-SY5Y [28] and nerve-like cells [40].

The present work was aimed at evaluating the effect of acute or chronic effects of ROT on the high yield of functional DALNs concerning OS, autophagy, and apoptosis. To achieve this, MenSC-derived DALNs were exposed to ROT (50 μM) for 6 h (acute) or ROT (10 μM) for 24 h (chronic). DALNs were then analyzed for ROT-induced mitochondrial damage according to ΔΨ_m_ assay. Additionally, we evaluated the translocation/colocalization of dynamin-related protein 1 (DRP1), which is implicated as a mediator of mitochondrial fission and apoptosis [41,42], with the translocase of the outer membrane of mitochondria 20 (TOMM20) protein. As OS evidence, the appearance of DJ-1Cys^106^-SO_3_ was examined. The presence of phosphorylated LRRK2 kinase at residue Ser^935^ and α-Syn at residue Ser^129^ was evaluated as a sign of the activation of proapoptotic LRRK2 kinase and pathological markers found in postmortem brains of PD patients [43]. We used the accumulation of lipidated microtubule-associated protein 1B-light chain 3 (LC3B) as a highly specific marker of autophagosomes, and Lysotraker^®^ to monitor lysosomes. Lastly, we evaluated cleaved caspase 3 (CC3) as a marker of activated caspase 3 (CASP3) in apoptotic cells. Our data suggest that apoptotic cell death ensues in both acute and chronic ROT conditions independently of the accumulation of LC3II and lysosomes. Given that defects in mitochondrial complex I activity are commonly observed in PD [44], ROT works well as a chemical model of PD in both acute and chronic conditions [45,46]. Therefore, prevention and treatment therapy should be guided to relieve DALNs from mitochondrial damage [47,48] and OS [49,50].

## 2. Results

### 2.1. Obtention of a High Yield of Menstrual Stromal Cell-Derived Mature and Functional Dopamine-like Neurons (DALNs) Using the NeuroForsk 2.0 Medium

Previously, it has been shown that MenSC-derived DALNs obtained in NeuroForsk medium yielded 26% TH+ and DAT+ neurons assessed by flow cytometry [7]. We tested whether adding growth factors FGFb and BDNF, the hormone melatonin, and the purine derivative small-molecule purmorphamine to NeuroForsk medium, now called NeuroForsk 2.0 medium, might increase the production of mature DALNs. To achieve this, MenSCs were cultured in NeuroForsk medium and NeuroForsk 2.0 medium. For comparative purposes, cells were cultured in RCm and maintenance medium NB B27^TM^. Figure 1 shows that MenSCs cultured in RCm and NB B27^TM^ medium expressed high basal percentages of neuronal lineage marker neurofilament light chain protein (NFL), a specific neuronal cytoskeleton protein, by 32% and 34%, respectively (Figure 1A,B). In contrast, MenSCs cultured in either NeuroForsk or NeuroForsk 2.0 medium expressed NFL marker (Figure 1A), albeit with different strength. The MenSC-derived DALNs increased the expression of NFL by +56%, whereas cells cultured in NeuroForsk 2.0 medium increased the expression of NFL by +147% in ChLNs compared to cells cultured in RCm (Figure 1A,B). Fluorescent microscopy analysis reproduced similar observations (Figure 1C,D). On the other hand, MenSC-derived DALNs expressed low basal levels of DAergic neuron markers TH and DAT in RCm and NB B27 TM medium (Figure 1E,F). While the percentage of DALNs, i.e., TH+/DAT+ cells increased by about +140% in NeuroForsk medium compared to RCm, the DALNs TH+/DAT+ cells increased by +590% in NeuroForsk 2.0 medium compared to RCm (Figure 1E,F). Of note, MenSCs cultured in NeuroForsk 2.0 medium increased DALNs TH+/DAT+ by +188% compared to MenSCs cultured in NeuroForsk medium. There was a statistically significant difference between RCm, NB B27^TM^, and NeuroForsk medium versus NeuroForsk 2.0 (Figure 1E). Similar observations were obtained by fluorescent microscopy (FM, Figure 1G–I).

The above observations prompted us to further evaluate the functional response of MenSC-derived DALNs to dopamine (DA) stimuli in different culture media. To achieve this aim, the cytoplasmic Ca^2+^ accumulation in DALNs was evaluated with Fluo-3AM-mediated Ca^2+^ imaging. While DALNs were unaffected in RCm and NB B27 medium (Figure 2A,B), DA induced a transient intracellular Ca^2+^ elevation under NeuroForsk and NeuroForsk 2.0 medium. Interestingly, while the maximal fluorescence change (ΔF/F) with NeuroForsk medium was 2.06 ± 0.03-fold after 10 s of DA exposure compared to cells (*p* < 0.05) (Figure 2B), the maximal fluorescence change (ΔF/F) in NeuroForsk 2.0 was 8.18 ± 0.3-fold at 10 s and 9.6 ± 0.4-fold at 80 s of DA addition (*p* < 0.05) (Figure 2B, blue broken lines). Based on the above and these observations, we therefore selected NeuroForsk 2.0 as an optimal culture medium for obtaining functional DALNs from MenSCs.

### 2.2. Acute Rotenone (ROT) Induces a Higher Loss of Mitochondrial Membrane Potential (ΔΨm), Expression of Parkin (PRKN) Protein, and Colocalization of DRP1 with Mitochondrial TOMM20 Protein than the Chronic Conditions in DALNs

Given that the primary target of ROT is mitochondrial complex I [35], we initially evaluated the effect of acute and chronic ROT exposure on ΔΨm in MenSC-derived DALNs. Figure 3A shows that acute or chronic ROT exposure induces loss of ΔΨm in DALNs, albeit with different strengths. While ROT provoked a reduction in ΔΨm by 49% in acute conditions (Figure 3B), a moderate decrease in ΔΨm by 13% was observed in chronic conditions. Effectively, ROT (50 μM) exposure for 6 h induced an important reduction in ΔΨm by 50% in DALNs when compared to cells exposed to ROT (10 μM) for 24 h. We also wanted to evaluated whether ROT affects PRKN, a cytosolic ring-between-ring E3 ligase protein involved in mitochondria quality control and apoptosis [51]. As shown in Figure 3C, acute ROT exposure increased the expression of PRKN by 514% in DALNs, whereas chronic conditions induced much lower expression of PRKN (+363%, Figure 3C,D) compared to untreated cells. When compared to chronic conditions, the acute ROT exposure yielded +16% PRKN expression. Regarding fluorescent microscopy analysis, similar observations were obtained for ΔΨm (Figure 3E,F) and PRKN (Figure 3G,H) in DALNs.

To further characterize the effect of ROT on mitochondria, we evaluated the cellular localization of the protein dynamin-related protein 1 (DRP1), which is implicated as a mediator of mitochondrial fission and apoptosis [41,42], via phosphorylation of LRRK2 [52]. Therefore, we assessed whether DRP1 colocalized with the translocase of the outer membrane of mitochondria 20 (TOMM20) was indicative of the cytoplasmic DRP1 translocation to mitochondria in DALNs under ROT treatment. As shown in Figure 4A, DRP1 displayed a cytoplasmic localization in untreated (0 μM condition) DALNs for 6 h, whereas it mostly colocalized with mitochondrial TOMM20 protein in DALNs (85%) when exposed to ROT (50 μM, acute conditions) (Merge and Figure 4B). Similarly, no colocalization of DRP1 and TOMM20 was observed in untreated DALNs cultured for 24 h (Figure 4C), but it colocalized with TOMM20 (50%) in DALNs when exposed to ROT (10 μM, chronic conditions). There was not only a significant difference between the untreated versus treated conditions on DALNs (Figure 4B or Figure 4D) but also a significant statistically difference between DALNs exposed to ROT (50 μM, 6 h) and ROT (10 μM, 24 h) in DALNs (Figure 4B,D, *p* = 0.003).

### 2.3. Acute Rotenone (ROT) Induces a Higher Oxidation of Protein DJ-1, Phosphorylation of LRRK2 Kinase, and Phosphorylation of α-Synuclein Protein Than Chronic Conditions in DALNs

Several lines of evidence suggest that ROT produces ROS [36], such as H_2_O_2_, which in turn acts as a second messenger, triggering the oxidation of proteins such as DJ-1 [53] and the phosphorylation of kinases (e.g., LRRK2 [54]). We wanted, therefore, to determine whether there is a differential effect of ROT concerning the oxidation of DJ-1 and phosphorylation of LRRK2 in DALNs under acute or chronic exposure. To achieve this, untreated or treated DALNs with ROT at either 50 μM or 10 μM for 6 h (acute) and 24 h (chronic), respectively, were analyzed for the presence of DJ-1Cys^106^-SO_3_ (-sulfhydryl group), indicative of OS, and phosphorylation of LRRK2 at Ser^935^, indicative of activation of the kinase. Flow cytometry analysis shows that under the acute ROT condition DALNs presented much more oxidized DJ-1 (Figure 5) and phosphorylated LRKK2 (Figure 6) than under the chronic ROT condition (Figure 5 and Figure 6). Indeed, acute ROT exposure increased oxDJ-1, i.e., DJ-1Cys^106^-SO_3_ by 1300% (Figure 5A,B) and pS^935^-LRRK2 by 750% (Figure 6A,B), whereas under the chronic ROT conditions, DJ-1Cys^106^-SO_3_ and pS^935^-LRRK2 were augmented by 583% and 266%, respectively (Figure 5A,B and Figure 6A,B). Total LRRK2 was similarly expressed in untreated cells as well as cells treated with ROT in acute and chronic exposure (Figure 6C,D). Of note, the acute exposure induced an important rise in DJ-1Cys^106^-SO_3_ and pS^935^-LRRK2 of 71% and 210%, respectively, when compared to cells exposed to chronic conditions. Similar data were obtained by fluorescent microscopy (Figure 5C,D and Figure 6E–G). Given that *p*-LRRK2 kinase phosphorylated α-Syn at Ser^129^ in HEK-293 cells [55], we evaluated whether ROT would be differentially phosphorylated α-Syn in DANLs under acute or chronic conditions. Thus, untreated or treated DALNs with ROT were analyzed. Acute ROT exposure induced a significant increase in pSer^129^-α-Syn (550%), whereas chronic ROT exposure provoked a much lower pSer^129^-α-Syn (+183%, Figure 6H,I). Acute conditions demonstrate that ROT (50 μM) raised pSer^129^-α-Syn by 53% compared to chronic exposure of DALNs to ROT (10 μM). The expression of total naïve α-Syn was not affected by ROT (Figure 6J,K). Similar observations were obtained by fluorescence microscopy (Figure 6L–N).

### 2.4. Acute Rotenone (ROT) Induces a Higher Activation of Caspase 3 (CASP3) Than the Chronic Conditions in DALNs

Cleaved caspase 3 (CC3) is considered a reliable marker for cells that are dying or have died by apoptosis [56]. Flow cytometry of CC3 showed that the percentage of CC3+ cells was increased in both acute and chronic conditions, although the acute condition induced a higher percentage of CC3+ (+1500%, Figure 7A,B) than the chronic condition (+500%). Acute conditions demonstrate that ROT (50 μM) raised CC3+ cells by +78% compared to chronic exposure of DALNs to ROT (10 μM). Similar data were obtained by fluorescence microscopy (Figure 7C,D).

### 2.5. Chronic Rotenone (ROT) Induces a Significant Accumulation of Lipidated LC3B, Lysosomes, and Autophagolysosomes in DALNs

Next, we evaluated whether ROT-induced accumulation of autophagosomes and autolysosomes relied on dose or time of exposure in DALNs. Microtubule-associated protein 1B light chain 3II (LC3II), a standard marker for autophagosomes, is generated by the conjugation of cytosolic LC3I with phosphatidylethanolamine (PE) on the surface of nascent autophagosomes [57]. As LC3II is relatively specifically associated with autophagosomes and autolysosomes (=autophagosome + lysosome), quantification of lipidated LC3B-positive cells is considered a gold-standard assay for monitoring cell organelle autophagy [58]. Flow cytometry analysis shows that acute ROT exposure induced an important buildup of lipidated LC3B+ cells by +460% (Figure 8A) and chronic ROT exposure brought a lower accrual of LC3B+ by +400% (Figure 8B). However, chronic conditions expressed much more LC3B+ (+30%) compared to acute conditions. As expected, rapamycin (RAP), used as a standard inducer of autophagy, induced the accumulation of LC3B+ cells by 38% (Figure 8C), whereas bafilomycin (BAF), used as an inhibitor of autophagy, induced high expression of LC3B+ cells by 363% (Figure 8D,E).

In addition to the above, we assessed whether acute or chronic ROT exposure induces the accumulation of lysosomes in DALNs. To achieve this aim, we used Lysotracker^®^, a cell-permeable, non-fixable, green fluorescent dye that stains acidic compartments within a cell, such as lysosomes. As shown in Figure 9, untreated DALNs showed basal acidic lysosome detection, i.e., lysotracker green-positive cells (Figure 9A, 2.56 AU and Figure 9B, 2.60 AU), whereas acute and chronic ROT exposure induced a statistically significant increase in lysosome detection by 104% Lysotrackers^®^+ cells (Figure 9A,E) and 146% Lysotrackers^®^+ cells (Figure 9B,E), respectively. The effect of the inducer of autophagy RAP was much lower (+27% Lysotrackers^®^ + cells, Figure 9C,E) than the chronic conditions (at 24 h) in DALNs, but the inhibitor BAF induced a comparable increase (154% Lysotrackers^®^+ cells, Figure 9D,E) to the chronic conditions.

### 2.6. The Inhibitor LRRK2 PF-06447475 (PF-475) Inhibits p-S^935^-LRKK2 and Cleaved Caspase 3 in DALNs Treated with ROT

To evaluate the effect of the inhibitor LRRK2 PF-06447475 (PF-475) on apoptotic cell death in acute or chronic ROT exposure conditions, DALNs were left untreated or treated with ROT in the presence or absence of PF-475. We first verified that PF-475 inhibited the phosphorylation of LRRK2. Effectively, PF-475 almost completely blocked the p-S^935^-LRRK2 in DALNs treated with ROT (e.g., −93%) compared to cells treated with ROT only at 6 h (Figure 10A,B) without affecting total LRRK2 (100%, Figure 10C,D). Similar results were evidenced when cells were exposed to chronic conditions (−60%, Figure 10E–H). In parallel, analysis of CC3 showed that PF-475 reduced the percentage of CC3-positive cells by 80% and 70% in acute (Figure 10I,J) and chronic conditions (Figure 10K,L), respectively, compared to cells treated with 50 μM or 10 μM ROT only for 6 h or 24 h, respectively (Figure 10I–L).

### 2.7. The Inhibitor LRRK2 PF-06447475 (PF-475) Reduces the Accumulation of Lysosomes in DALNs Treated with ROT

We further investigated whether PF-475 affects the lysosome pathway in DALN cells. Then, cells were left untreated or treated with ROT in the presence or absence of PF-475. Figure 11 shows that PF-475 reduced the accumulation of lysosomes in DALNs exposed to acute (−25%, Figure 11A,B) and chronic ROT (−45%, Figure 11C,D) compared to ROT only according to lysotracker^®^ assay (Figure 10B,D).

## 3. Discussion

Parkinson’s disease (PD) is characterized by the loss of DAergic neurons in the substantia nigra pars compacta, for which the etiology is not yet entirely established. Unfortunately, the disorder remains incurable. This is in part due to the lack of an experimental disease model. Therefore, the development of protocols to culture either hPSCs [8] or MSCs (e.g., [59]) differentiated into DA neurons has allowed the study of molecular aspects of PD (e.g., [40,60]). Previous studies have shown that MenSC-derived DALNs obtained in NeuroForsk medium yielded a modest 26% TH+ and DAT+ neurons [7]. The major component of this medium is forskolin, a cAMP-elevating agent [61] that induces neural-like differentiation of MSCs via downregulation of expression of the master transcriptional regulator neuron restrictive silencer factor (NRSF) and its downstream target genes [62]. This suggests that the adenyl-cyclase cAMP-elevating agent forskolin is necessary but insufficient in the transdifferentiation of MenSCs into DALNs. Several signaling molecules work closely together to program MenSCs to adopt DAergic neuronal fates (Figure 1 in [63]). In this study, we report for the first time that MenSCs cultured in NeuroForsk 2.0 medium transdifferentiated into DALNs by 70% TH+ and DAT+ cells as assessed by flow cytometry analysis. Supplementation of NeuroForsk with the most commonly reported factors, such as FGFb, BDNF, the hormone melatonin, and a synthetic agonist of smoothened purmorphamine, resulted in a new brand of NeuroForsk 2.0 medium. Indeed, FGFb and BDNF have a crucial role in survival and terminal differentiation of neuronal populations during development [64,65], whereas purmorphamine activates the Hedgehog (Hh) pathway by targeting Smoothened, a critical component of the Hh signaling pathway [66]. In fact, Hh signaling is an important regulator of embryonic patterning, tissue regeneration, and stem cell renewal [67], and melatonin enhances neural stem cell differentiation and engraftment by increasing mitochondrial function [68]. Compared with other protocols, wherein the yield of TH+ DA neurons derived from hPSCs was between 50% and 85% [9,10,11] or TH+ DA neurons derived from hMSCs was between 23% and 59% [12,13], the production of ~70% TH+/DAT+ DALNs from MenSCs in 7 days makes NeuroForsk 2.0 a time-saving, economical, and reliable medium to harness DALNs. Furthermore, we found that, compared to regular culture medium (RCm), NeuroForsk 2.0 medium yielded 590% TH+/DAT+ DALNs, whereas NeuroForsk generated TH+/DAT+ cells by 140%. Neurobasal (NB) B27^TM^ medium produced (<10%) DALNs to a comparable extent as RCm. Most interestingly, transdifferentiated DALNs in NeuroForsk 2.0 medium responded to DA stimuli by displaying two maximal fluorescence changes (ΔF/F) of transient Ca^2+^ influx. This observation suggests that DALNs are functional, most probably involving D2 autoreceptors and Ca^2+^ influx via L-type voltage channels [69]. Remarkably, MenSCs cultured in RCm and NB B27^TM^ medium did not respond to DA stimuli. Taken together, these observations suggest that the NeuroForsk 2.0 medium is highly efficient in producing specific, i.e., NFL-positive, cells and functional, i.e., Ca^2+^-responsive DALNs in vitro.

Previous studies have shown that ROT-induced apoptosis in neuron-like cells and non-neuronal cells (e.g., HEK-293 cells) occurs through OS mechanisms involving the loss of ΔΨ_m_, generation of H_2_O_2_, oxidation of DJ-1 into DJ-1Cys^106^-SO_3_, increased expression of PRKN, phosphorylated LRRK2 at residue Ser^935^, phosphorylated α-Syn at residue Ser^129^, mitochondrial colocalization of DRP1 and TOMM20, and appearance of cleaved caspase 3 (CC3) as a sign of the activation of CASP3 and apoptosis [40,70,71]. Here, we confirm that ROT induced a dose (10 and 50 μM)- and time (6 and 24 h)-dependent OS mechanism in DALNs involving positive detection of all OS and apoptotic markers such as ΔΨ_m_, DJ-1Cys^106^-SO_3_/(H_2_O_2_), DRP1/TOMM20 and PRKN, p-Ser^935^-LRRK2, p-Ser^129^-α-Syn and CC3 (Table 1). Clearly, acute conditions (ROT 50 μM, 6 h) induce apoptosis in DALNs more drastically than chronic conditions (ROT 10 μM, 24 h). Despite this differential effect (Table 1, acute versus chronic), acute or chronic ROT exposure triggers a similar molecular mechanism in DALNs.

For completeness, we further investigated the effects of both conditions on autophagy. ROT causes upregulation of lipidated LC3B and the accumulation of autophagosomes and lysosomes in DALNs. These data demonstrate the involvement of apoptosis and autophagy in the ROT-induced parkinsonian model in vitro. However, such effects were greater for the chronic conditions than for the acute conditions (Table 1). Interestingly, the higher total cleavage of CASP3 (CC3) in the acute conditions suggests that the apoptotic pathway is a more rapid process than the autophagy pathway. Indeed, in the context of ROT intoxication, autophagy might be viewed as a signature of ROT action, but it is time-dependent. Several observations support this assumption. First, acute ROT exposure induces a significant rise in OS (i.e., oxidized DJ-1Cys^106^-SO_3_), mitochondria damage (i.e., loss of ΔΨ_m_), and cell death (up-generation of CC3) markers compared to chronic ROT exposure in DALNs. Second, autophagy was higher in chronic than acute conditions without an apparent increase in the apoptotic marker CC3. Third, it has been demonstrated that ROT not only targets mitochondria but also inhibits the lysosomal glucosylceramidase beta 1 (GBA1) enzyme, leading to the accumulation of lysosomes and autophagosomes independently of apoptosis [31]. Interestingly, a similar result was obtained with the specific inhibitor GBA1 conduritol-B-epoxide [31]. Four, adaptive autophagy (as it may be the case in the ROT paradigm) might not be an important contributing factor to cell death because most probably it mediates cytoprotective rather than cytotoxic effects on DALNs exposed to ROT [72,73]. Fifth, no autophagy-dependent cell death has yet been demonstrated for ROT intoxication. Rather, studies involving genetic manipulation of autophagy in physiological settings provide evidence for a direct role of autophagy in specific scenarios [74], e.g., autosis in HeLa cells [75]. Therefore, autophagy can also act as a pro-death process in a context-dependent manner. Last, antioxidant agents almost completely abrogate ROT-induced OS and cell death in vitro and in vivo [76]. Accordingly, the antioxidant and protective effects underrate the contribution of autophagy in ROT-induced cell death. Taken together, these observations suggest that despite impairment of the autophagy pathway, it does not contribute to ROT-induced apoptosis in DALNs. Moreover, the induction of autophagy by ROT coincides with the induction of apoptosis in DALNs, where autophagy simply accompanies the cell death process, but does not have an active role in it. Interestingly, the inhibitor PF-475 efficiently blocked apoptosis and reduced accumulation of lysosomes in DALNs treated with ROT. These observations suggest that LRRK2 is a master kinase involved in the regulation of apoptosis and autophagy in DALNs under OS. Therefore, development of chemical inhibitors of LRRK2 might be an efficacious disease-modifying therapy for PD.

## 4. Materials and Methods

### 4.1. Source of Menstrual Stromal Cells (MenSCs)

The menstrual stromal cells (MenSCs) were obtained from the Neuroscience Tissue Bank (NTB) under code number 69308 (University of Antioquia, UdeA).

### 4.2. Dopaminergic-like Neurons (DALNs) Differentiation

The MenSCs were seeded at 1 × 10^4^ MenSCs/cm^2^ in 25 cm^2^ culture flasks for 24 h in regular culture medium (RCm, low-glucose DMEM supplemented with 10% FBS). Then, the medium was removed and cells were incubated either in Neurobasal B-27 medium (NB-B27, 2%), NeuroForsk medium (low-glucose DMEM supplemented with 2% FBS, and Forskolin (Sigma cat# F6886), 1 μM final concentration), or NeuroForsk 2.0 medium (FGFb 20 ng/μL, BDNF 50 ng/mL, melatonin 1 μM, purmorphamine 250 ng/mL, and Forskolin 1 μM) for 7 days.

### 4.3. Immunofluorescence Analysis

For immunofluorescence analysis of dopamine (DA) neuron markers, cells treated with RCm, NB-B27, NeuroForsk, or NeuroForsk 2.0 medium for 7 days were fixed with paraformaldehyde for 20 min, followed by triton X-100 (0.1%) permeabilization and 5% bovine serum albumin (BSA) blockage. Cells were then incubated overnight with primary antibodies against DAT and TH proteins (1:200). After exhaustive rinsing, we incubated the cells with secondary fluorescent antibodies (DyLight 488 and 595 donkey anti-rabbit, -goat, and -mouse, Cat DI 2488, and DI 1094, respectively, 1:500). The nuclei were stained with Hoechst 33342 (1 µM, Life Technologies) and images were acquired on a Floyd Cells Imaging Station microscope.

### 4.4. Flow Cytometry Analysis of Dopaminergic Markers

Flow cytometry acquisition was used to determine the percentage of NFL and DAT/TH double-positive cells based on previous reports [77,78]. Cells cultured in RCm, NB-B27^TM^, NeuroForsk, or NeuroForsk 2.0 medium at day 7 were detached with 0.25% trypsin and 1 mM EDTA and fixed in suspension with paraformaldehyde overnight. After washing, cells were simultaneously incubated with DAT and TH primary antibodies (1:200) at 4 °C overnight. Cell suspensions were washed and incubated with DyeLight 594 donkey anti-goat and DyeLight 488 donkey anti-rabbit antibodies (1:500). Finally, cells were washed and resuspended in PBS for analysis on a Canto cytometer (Beckman Coulter). Ten thousand events were acquired, and the acquisition analysis was performed using FlowJo 7.6.2 Data Analysis Software. Positive staining was defined as the fluorescence emission that exceeded levels in the population stained with the negative control (only secondary antibody staining).

### 4.5. Intracellular Calcium Imaging

The cytoplasmic Ca^2+^ concentration ([Ca^2+^]_i_) was measured according to [79,80]. Briefly, DALNs cultured in RCm, NB-B27, NeuroForsk, and NeuroForsk 2.0 medium, respectively, for 7 days were transferred to a bath solution (NBS; in mM: 137 NaCl, 5 KCl, 2.5 CaCl_2_, 1 MgCl_2_, 10 HEPES, pH 7.3, and 22 glucose) containing a Ca^2+^-sensitive indicator (2 µM Fluo3-AM, an acetoxymethyl ester form of the fluorescent dye Fluo-3; Thermo Fisher Scientific Cat F1242) for 30 min at room temperature and then washed five times. The intracellular Ca^2+^ transients were evoked by dopamine hydrochloride (DA, 1 mM final). The amplitudes of the Ca^2+^-related fluorescence transients were expressed relative to the resting fluorescence (ΔF/F) and were calculated by the formula ΔF/F = (F_max_ − F_rest_)/(F_rest_ − F_bg_) [79]. The Image J program (https://imagej.net/, accessed on 1 June 2023) was used for the calculation of the fluorescence intensities.

### 4.6. Detection of oxDJ-1, Phosphorylated LRRK2, Alpha-Synuclein, Cleaved Caspase 3 (CC3), and LC3B Using Fluorescent Microscopy or Flow Cytometry

After treatment in the absence or presence of ROT (10, 50 μM) with or without inhibitor LRRK2 PF-06447475 (1 μM) for 6 or 24 h, cells (1 × 10^5^) were fixed in 80% ethanol and stored at 20 °C overnight. Then, cells were washed with PBS and permeabilized with 0.2% triton X-100 (Cat# 93443, Sigma-Aldrich, St. Louis, MO, USA) plus 1.5% bovine serum albumin (BSA, Cat# A9418, Sigma-Aldrich, St. Louis, MO, USA) in phosphate-buffered solution (PBS) for 30 min. Then, cells were washed and incubated with primary antibodies (1:200; diluted in PBS containing 0.1% BSA) against oxidized DJ-1 (1:500; ox (Cys106) DJ1; spanning residue C106 of human PARK7/DJ1; oxidized to produce cysteine sulfonic (SO3) acid; Abcam cat# AB169520; Boston, MA, USA), cleaved caspase 3, (CC3; 1:250; cat# AB3623, Millipore, Merck, Darmstadt, Germany), p-(S935)-LRRK2 (Abcam cat# AB133450; Boston, MA, USA), α-synuclein (pS129; Abcam cat# AB51253; Boston, MA, USA), and LC3B (cat# NB100-2220, Novus Biologicals, Englewood, CO, USA) overnight at 4 °C. After exhaustive rinsing, we incubated the cells with secondary fluorescent antibodies (DyeLight 488 horse anti-rabbit and -mouse antibodies, cats DI 1094 and DI 2488, Vector Laboratories, Newark, NJ, USA) at 1:500. Finally, cells were washed and resuspended in PBS for analysis on a BD LSRFortessa II flow cytometer (BD Biosciences, Franklin Lakes, NJ, USA). Twenty thousand events were acquired, and the acquisition analysis was performed using FlowJo 7.6.2 Data Analysis Software (BD Biosciences, Franklin Lakes, NJ, USA). For microscopy, the nuclei were stained with 0.5 μM Hoechst 33342 (Life Technologies, Carlsbad, CA, USA), and images were acquired on a Floyd Cells Imaging Station microscope (Cat# 4471136, Life Technologies, Carlsbad, CA, USA).

### 4.7. Analysis of Mitochondrial Membrane Potential (ΔΨm)

The assessment of ΔΨm was performed according to [81]. Briefly, cells were incubated for 20 min at RT in the dark with a deep-red MitoTracker (20 nM final concentration) compound (cat# M22426, Thermo Scientific, Waltham, MA, USA). Cells were analyzed using a fluorescence microscopy Floid Cells Imaging Station microscope (cat# 4471136, Life Technologies, Carlsbad, CA, USA) or a BD LSRFortessa II flow cytometer (BD Biosciences, Franklin Lakes, NJ, USA). The experiment was conducted three times, and 10,000 events were acquired for analysis. Highly positive MitoTracker cells were selected located between 104 and 106. No discrimination by complexity was made. Quantitative data and figures were obtained using FlowJo 7.6.2 Data Analysis Software (BD Biosciences, Franklin Lakes, NJ, USA).

### 4.8. Characterization of Lysosomal Complexity

To analyze lysosomal complexity, untreated or treated cells were incubated with the cell-permeable, non-fixable, green fluorescent dye LysoTracker Green DND-26 (50 nM, cat# L7526, Thermo Fisher Scientific, Waltham, MA, USA) for 30 min at 37 °C. Cells were then washed, and LysoTracker fluorescence was determined by analysis of fluorescence microscopy images in a Floid Cells Imaging Station microscope (Cat# 4471136, Life Technologies, Carlsbad, CA, USA) or flow cytometry using a BD LSRFortessa II flow cytometer (BD Biosciences, Franklin Lakes, NJ, USA). The experiment was conducted three times, and 10,000 events were acquired for analysis. Flow cytometry analysis for LysoTracker/SSCA was performed by selecting, in the FL-1 channel, all cells with LysoTracker reactivity (>99%), in order to perform the analysis of the total LysoTracker-positive population. The SSCA parameter was adjusted to the mean fluorescence of the control (UNT: 40 K ± 3.5 K) plus two standard deviations (i.e., values above 47 K). Quantitative data and figures were obtained using FlowJo 7.6.2 Data Analysis Software (BD Biosciences, Franklin Lakes, NJ, USA).

### 4.9. Data Analysis

In this experimental design, a vial of MenSCs was thawed and cultured, and the cell suspension was pipetted at a standardized cellular density of 2 × 10^4^ cells per cm^2^ into different wells of a 24-well plate. Cells (i.e., the biological and observational units) were randomized to wells by simple randomization (sampling without replacement method), and then wells (i.e., the experimental units) were randomized to treatments by a similar method. Experiments were conducted in triplicate. The data from individual replicate wells were averaged to yield a value of *n* = 1 for that experiment, and this was repeated on three occasions blind to the experimenter and/or flow cytometer analyst for a final value of *n* = 3 [82]. Based on the assumptions that the experimental unit (i.e., the well) data comply with independence of observations, the dependent variable is normally distributed in each treatment group (Shapiro–Wilk test) and there is homogeneity of variance (Levene’s test). Statistical significance was determined by a one-way analysis of variance (ANOVA) followed by Tukey’s post hoc comparison calculated with GraphPad Prism 5.0 software. Differences between groups were only deemed significant when a *p*-value of 0.05 (*), 0.001 (**), or 0.001 (***) was obtained. All data are given as the mean ± SD.

## 5. Conclusions

We studied the NeuroForsk 2.0 medium, which is highly efficient in producing specific and functional DALNs (70% TH+/DAT+) in 7 days for in vitro experiments. NeuroForsk 2.0 should be the first-choice medium to obtain MenSC-derived DALNs in a time-saving, economical, and reliable protocol to harness DALNs. Concerning the effect of ROT, our data suggest that DALNs are more susceptible to undergo apoptosis than ROT-induced autophagy (Figure 12). The evidence gathered in the present investigation suggest that targeting apoptosis rather than autophagy is a potential treatment strategy for PD. Given that the pathogenesis of PD is complex, not fully understood, and the therapeutic options are limited, elucidating the cross talk (if any) between apoptosis and autophagy in PD is essential from an etiological standpoint and for developing effective therapeutic strategies.

## Figures and Tables

**Figure 1 ijms-24-15744-f001:**
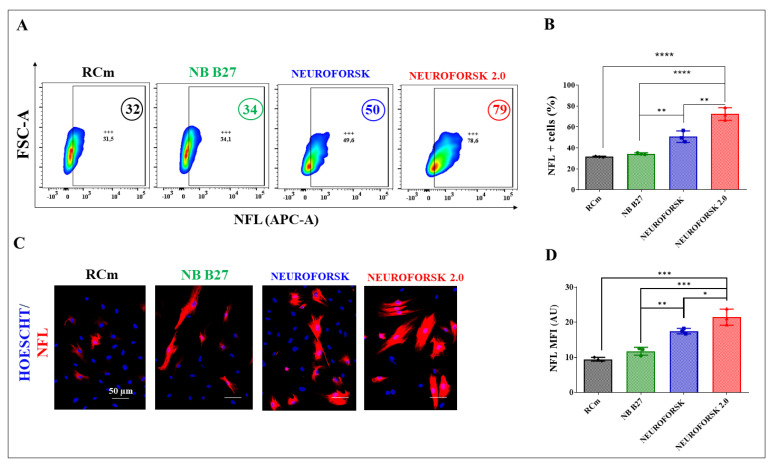
Immunofluorescence analysis, percentage expression of neuronal lineage and dopaminergic markers in MenSCs differentiated into dopamine-like neurons (DALNs). MenSCs were cultured in RCm, NB-B27, NeuroForsk or NeuroForsk 2.0 medium as described in the Section 4. After this time, cells were single- or double-stained as indicated in the figure with primary antibodies against neurofilament light chain (NFL), tyrosine hydroxylase (TH), and dopamine transporter (DAT). (**A**) Representative density plot figures showing the NFL of single-stained NFL in RCm, NB-B27, NeuroForsk, or NeuroForsk 2.0 medium cultured MenSCs at day 7. (**B**) Percentage of NFL-positive cells in RCm, NB-B27, NeuroForsk, and NeuroForsk 2.0 medium-cultured MenSCs at day 7. (**C**) Representative fluorescence microscopy photographs showing MenSCs cultured in RCm (black bar), NB-B27 (green bar), NeuroForsk (blue bar), and NeuroForsk 2.0 (red bar) incubated with primary antibodies against NFL (red) at day 7. The nuclei were stained with Hoechst. (**D**) Mean fluorescence intensity (MFI) quantification of images obtained by immunofluorescence analysis. (**E**) Representative density plot figures showing the TH/DAT double-positive population in RCm, NB-B27, NeuroForsk, or NeuroForsk 2.0 medium-cultured MenSCs at day 7. (**F**) Percentage of TH/DAT double-positive cells in RCm (black bar), NB-B27 (green bar), NeuroForsk (blue bar), and NeuroForsk 2.0 (red bar) medium-cultured MenSCs at day 7. (**G**) Representative fluorescence microscopy photographs showing MenSCs cultured in RCm, NB-B27, NeuroForsk, and NeuroForsk 2.0 medium incubated with primary antibodies against TH (green) and DAT (red) at day 7. The nuclei were stained with Hoechst. (**H**,**I**) Mean fluorescence intensity (MFI) quantification of images obtained by immunofluorescence analysis. The figures represent 1 out of 3 independent experiments. One-way ANOVA, post hoc test Bonferroni. Data are presented as mean ± SD (* *p* < 0.05; ** *p* < 0.01; *** *p* < 0.001). Image magnification, 200×. +++ symbols and numbers in the figure represent positive cellular population for the tested marker.

**Figure 2 ijms-24-15744-f002:**
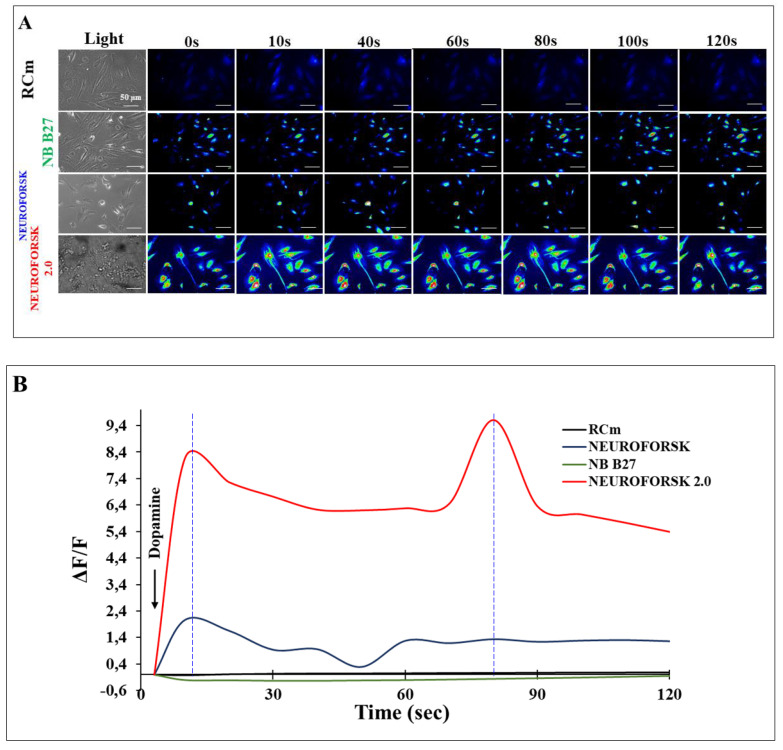
Evaluation of dopamine (DA) response in dopamine-like neurons (DALNs). (**A**) MenSCs were cultured in RCm, NB-B27, NeuroForsk and NeuroForsk 2.0 medium as described in Section 4. Time-lapse images (0, 10, 40, 60, 80, 100 and 120 s) of Ca^2+^ fluorescence in MenSC cultured in RCm, NB-B27, NeuroForsk, and NeuroForsk 2.0 medium at day 7 (*n* = 30 cells imaged, N = 3 dishes) as a response to dopamine (DA) treatment. DA was puffed into the culture at 0 s. Then, the Ca^2+^ fluorescence of cells was monitored at the indicated times. Color contrast indicates fluorescence intensity: dark blue < light blue < green < yellow < red. (**B**) Normalized mean fluorescence signal (ΔF/F) over time, indicating temporal cytoplasmic Ca^2+^ elevation in response to DA treatment. The figures represent 1 out of 3 independent experiments. Data are presented as mean (*n* = 3). Image magnification, 200×.

**Figure 3 ijms-24-15744-f003:**
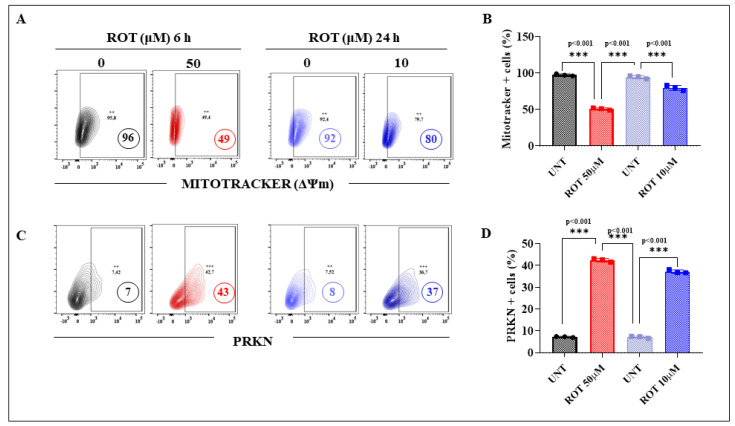
Acute and chronic exposure to rotenone induces significant ΔΨm damage and increased PRKN expression in DANLs. (**A**) Representative flow cytometry contour plots showing the percentage of MitoTracker of untreated DALNs or those treated with ROT (50 µM) for 6 h and untreated or those treated with ROT (10 µM) for 24 h. (**B**) Percentage of MitoTracker-positive cells. (**C**) Representative flow cytometry contour plots showing untreated DALNs or those treated with ROT (50 µM) for 6 h and untreated or those treated with ROT (10 µM) for 24 h and stained with primary antibody against parkin (PRKN). (**D**) Percentage of PRKN-positive cells. (**E**) Representative fluorescence microscopy photographs showing untreated DALNs or those treated with ROT (50 µM) for 6 h and untreated or those treated with ROT (10 µM) for 24 h, and stained with MitoTracker. Positive red fluorescence reflects high mitochondrial membrane potential ΔΨ_m_, and positive blue fluorescence reflects nuclei. (**F**) Quantification of the MitoTracker mean fluorescence intensity (MFI) in untreated and treated DANLs. (**G**) Representative fluorescence microscopy photographs showing untreated DALNs or those treated with ROT (50 µM) for 6 h and untreated or those treated with ROT (10 µM) for 24 h and stained with primary antibody against parkin (PRKN). Positive blue fluorescence reflects nuclei, positive green fluorescence reflects the presence of PRKN protein. (**H**) Quantification of the PRKN mean fluorescence intensity (MFI) in untreated and treated DANLs. The figures represent 1 out of 3 independent experiments. One-way ANOVA, followed by Tukey’s test. Statistically significant differences: ** *p* < 0.01, and *p* *** < 0.001. Image magnification, 200×. The contour diagrams, histograms, bars, dot graphs, and photomicrographs represent one out of three independent experiments (*n* = 3). The data are presented as mean ± SD of three independent experiments. SD represents <5%. ++, +++ symbols and numbers in the figure represent positive cellular population for the tested marker.

**Figure 4 ijms-24-15744-f004:**
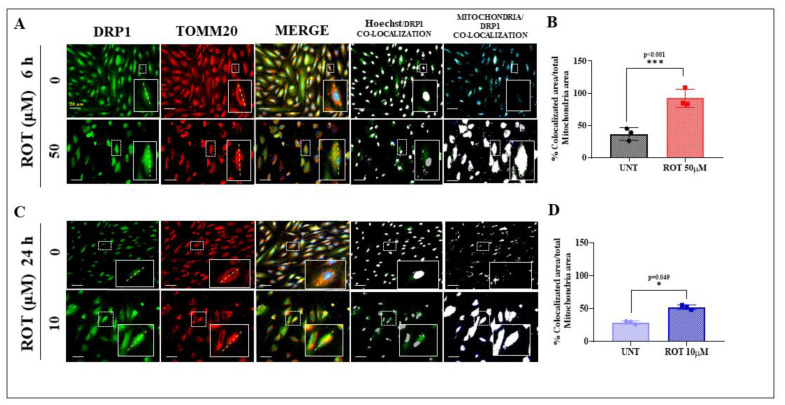
TOMM20 and DRP1 colocalize in DANLs cells exposed to acute and chronic ROT doses. (**A**) Representative fluorescence microscopy photographs showing untreated DALNs or those treated with (50 µM) ROT for 6, stained with primary antibodies against DRP1 (green fluorescence image) and TOMM20 (red fluorescence image). Image of colocalization of DRP1 and mitochondrial TOMM20 and merged image. Image of colocalization of nuclei (Hoechst) and DRP1. Image of colocalization of mitochondria and DRP1. (**B**) Percentage of localization area of colocalization of DRP1/mitochondria. (**C**) Representative fluorescence microscopy photographs showing untreated DALNs or those treated with 10 µM ROT for 24 h, stained with primary antibodies against DRP1 (green fluorescence image) and TOMM20 (red fluorescence image). Image of colocalization of DRP1 and mitochondrial TOMM20 and merged image. Image of colocalization of nuclei (Hoechst) and DRP1. Image of colocalization of mitochondria and DRP1. (**D**) Percentage of localization area of colocalization of DRP1/mitochondria. The image represents one out of three independent experiments (*n* = 3). One-way ANOVA test showed a statistically significant intragroup and intergroup (Tukey’s multiple comparison test, *p* = 0.003) differences. Statistically significant differences: * *p* < 0.05, and *p* *** < 0.001. The data are presented as mean ± SD of three independent experiments. Image magnification, 400×.

**Figure 5 ijms-24-15744-f005:**
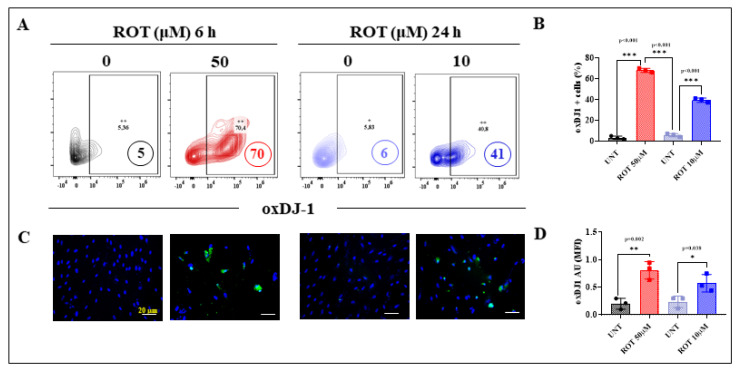
DALNs show a high oxidation of DJ-1 under acute but moderate under chronic ROT exposure. (**A**) Representative flow cytometry contour plots showing untreated DALNs or those treated with ROT (50 µM) for 6 h and untreated DALNs or those treated with ROT (10 µM) for 24 h, and stained with primary antibody against oxidized protein DJ-1 (DJ-1Cys^106^-SO_3_, oxDJ-1). (**B**) Percentage of DJ-1Cys^106^-SO_3_-positive cells. (**C**) Representative fluorescence microscopy photographs showing untreated DANLs or those treated with ROT (50 µM) for 6 h and untreated DANLs or those treated with ROT (10 µM) for 24 h and stained with Hoechst (blue) and primary antibodies against oxDJ-1 (green). Positive blue fluorescence reflects nuclei and positive green fluorescence reflects the presence of oxDJ-1. (**D**) Quantification of the oxDJ-1. The data are presented as the mean ± SD of three independent experiments (*n* = 3). One-way ANOVA followed by Tukey’s test: Statistically significant differences: * *p* < 0.05; ** *p* < 0.01, *** *p* < 0.001. Image magnification, 200×. +, ++ symbols and numbers in the figure represent positive cellular population for the tested marker.

**Figure 6 ijms-24-15744-f006:**
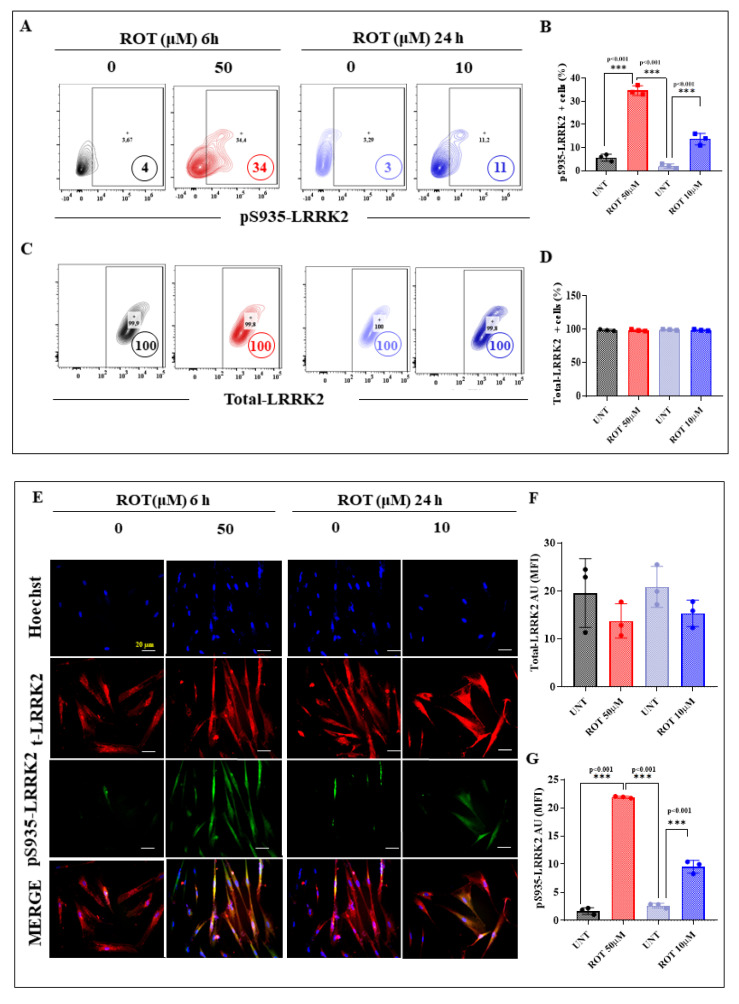
DALNs show a high oxidation of phosphorylation of LRRK2 and phosphorylation of α-Syn under acute but moderate under chronic ROT exposure. (**A**) Representative flow cytometry contour plots showing untreated DALNs or those treated with ROT (50 µM) for 6 h and untreated DALNs or those treated with ROT (10 µM) for 24 h, and stained with primary antibody against phosphorylated LRRK2 (pS^935^-LRRK2). (**B**) Percentage of pS^935^-LRRK2-positive cells. (**C**) Representative flow cytometry contour plots showing untreated DALNs or those treated with ROT (50 µM) for 6 h, and untreated DALNs or those treated with ROT (10 µM) for 24 h and stained with primary antibody against naïve LRRK2. (**D**) Percentage of LRRK2-positive cells. (**E**) Representative fluorescence microscopy photographs showing untreated DANLs or those treated with ROT (50 µM) for 6 h and untreated DANLs or those treated with ROT (10 µM) for 24 h and stained with Hoechst and primary antibodies against pS^935^-LRRK2 and naïve LRRK2. Positive blue fluorescence reflects nuclei, positive green fluorescence reflects the presence of pS^935^-LRRK2, and positive red fluorescence reflects t-LRRK2 protein. (**F**,**G**) Mean fluorescence intensity (MFI) in untreated or treated DANLs. (**H**) Representative flow cytometry contour plots showing untreated DALNs or those treated with ROT (50 µM) for 6 h and untreated DALNs or those treated with ROT (10 µM) for 24 h and stained with primary antibody against phosphorylated α-synuclein (pS^129^-αSYN). (**I**) Percentage of pS^129^-αSYN-positive cells. (**J**) Representative flow cytometry contour plots showing untreated DALNs or those treated with ROT (50 µM) for 6 h and untreated DALNs or those treated with ROT (10 µM) for 24 h and stained with primary antibody against naïve αSYN. (**K**) Percentage of total- αSYN-positive cells. (**L**) Representative fluorescence microscopy photographs showing untreated DANLs or those treated with ROT (50 µM) for 6 h and untreated DANLs or those treated with ROT (10 µM) for 24 h and stained with Hoechst and primary antibodies against pS^129^-αSYN and naïve αSYN. Positive blue fluorescence reflects nuclei, positive green fluorescence reflects the presence of pS^129^-αSYN, and positive red fluorescence reflects total αSYN protein. (**M**,**N**) Mean fluorescence intensity (MFI) in untreated or treated DANLs. The data are presented as the mean ± SD of three independent experiments (*n* = 3). One-way ANOVA followed by Tukey’s test. Statistically significant differences: * *p* < 0.05; *** *p* < 0.001. Image magnification, 200×. + symbol and numbers in the figure represent positive cellular population for the tested marker.

**Figure 7 ijms-24-15744-f007:**
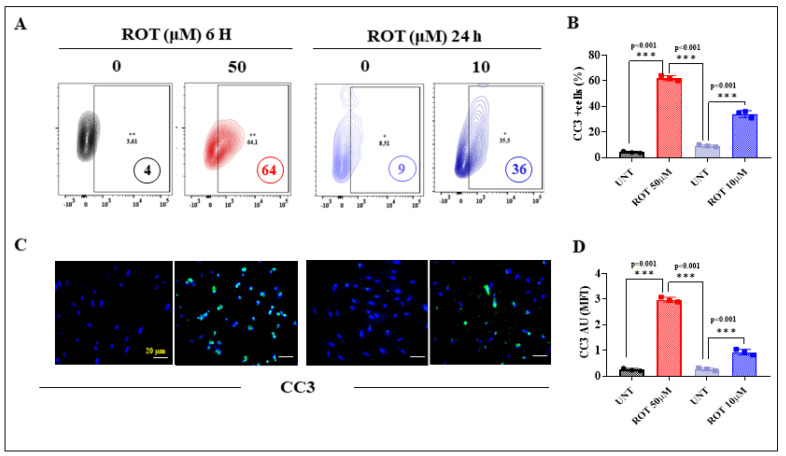
DALNs show high-oxidation cleaved caspase 3 (CC3) under acute and chronic ROT exposure. (**A**) Representative flow cytometry contour plots showing untreated DALNs or those treated with ROT (50 µM) for 6 h and untreated DALNs or those treated with ROT (10 µM) for 24 h and stained with primary antibody against cleaved caspase-3 (CC3). (**B**) Percentage of CC3-positive cells. (**C**) Representative fluorescence microscopy photographs showing untreated DANLs or those treated with ROT (50 µM) for 6 h and untreated DANLs or those treated with ROT (10 µM) for 24 h and stained with Hoechst and primary antibodies against protein CC3 (green). Positive blue fluorescence reflects nuclei, positive green fluorescence reflects the presence of CC3 protein. (**D**) Mean fluorescence intensity (MFI) in untreated or treated DANLs. The data are presented as the mean ± SD of three independent experiments (*n* = 3). One-way ANOVA followed by Tukey’s test: Statistically significant differences: *** *p* < 0.001. Image magnification, 200×. +, ++ symbols and numbers in the figure represent positive cellular population for the tested marker.

**Figure 8 ijms-24-15744-f008:**
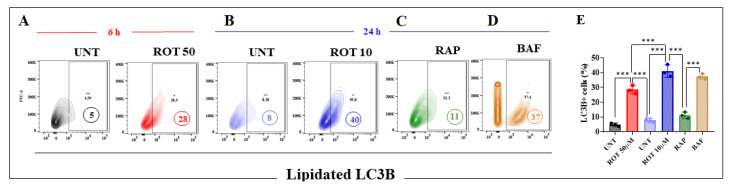
Chronic ROT exposure increases lipidated LC3B more than acute conditions in DALNs. (**A**) Representative contour 2D plots showing the percentage of lapidated LC3B of untreated DALNs or those treated with ROT (50 μM) for 6 h, (**B**) untreated DALNs or those treated with ROT (10 μM) for 24 h, (**C**) rapamycin (RAP, 10 nM), (**D**) bafilomycin A1 (BAF, 10 nM) for 24 h. (**E**) Percentage of LC3B-positive cells. The data are presented as the mean ± SD of three independent experiments (*n* = 3). One-way ANOVA followed by Tukey’s test. Statistically significant differences: *** *p* < 0.001. +, ++ symbols and numbers in the figure represent positive cellular population for the tested marker.

**Figure 9 ijms-24-15744-f009:**
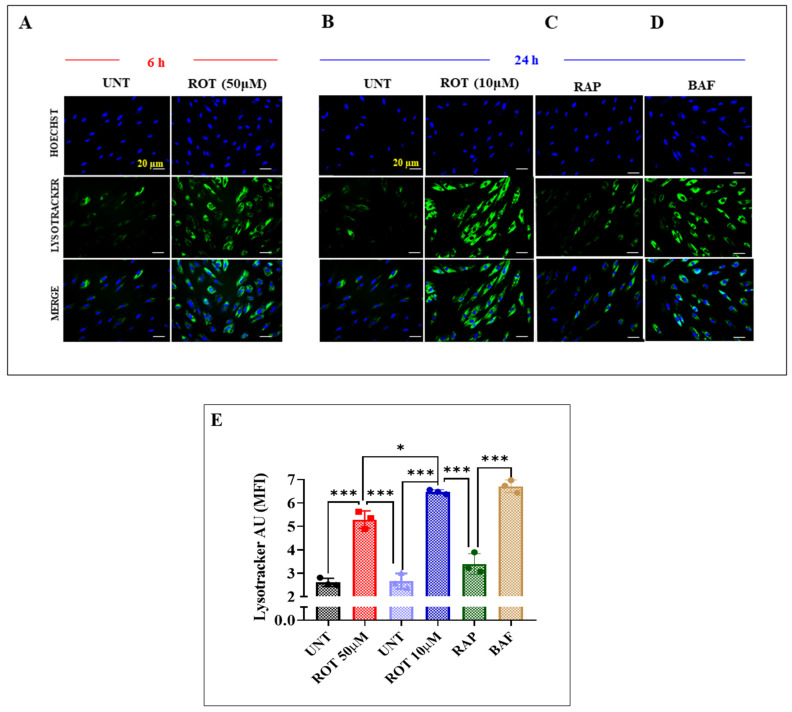
Chronic rotenone exposure induces a higher accumulation of lysosomes than acute conditions in DALN. (**A**) Representative immunofluorescence images showing lysosomal accumulation by Lysotracker^®^ stain in untreated DANLs or those treated with ROT (50 μM) for 6 h; (**B**) untreated DANLs or those treated with ROT (10 μM), (**C**) rapamycin (RAP, 10 nM), and (**D**) bafilomycin A1 (BAF, 10 nM) for 24 h. Positive blue fluorescence reflects nuclei, positive green fluorescence reflects the Lysotracker stain. (**E**) Quantitative analysis of Lysotracker^®^ accumulation as mean fluorescence intensity. The data are presented as the mean ± SD of three independent experiments (*n* = 3). One-way ANOVA followed by Tukey’s test. Statistically significant differences: * *p* < 0.05, *** *p* < 0.001. Image magnification, 200×.

**Figure 10 ijms-24-15744-f010:**
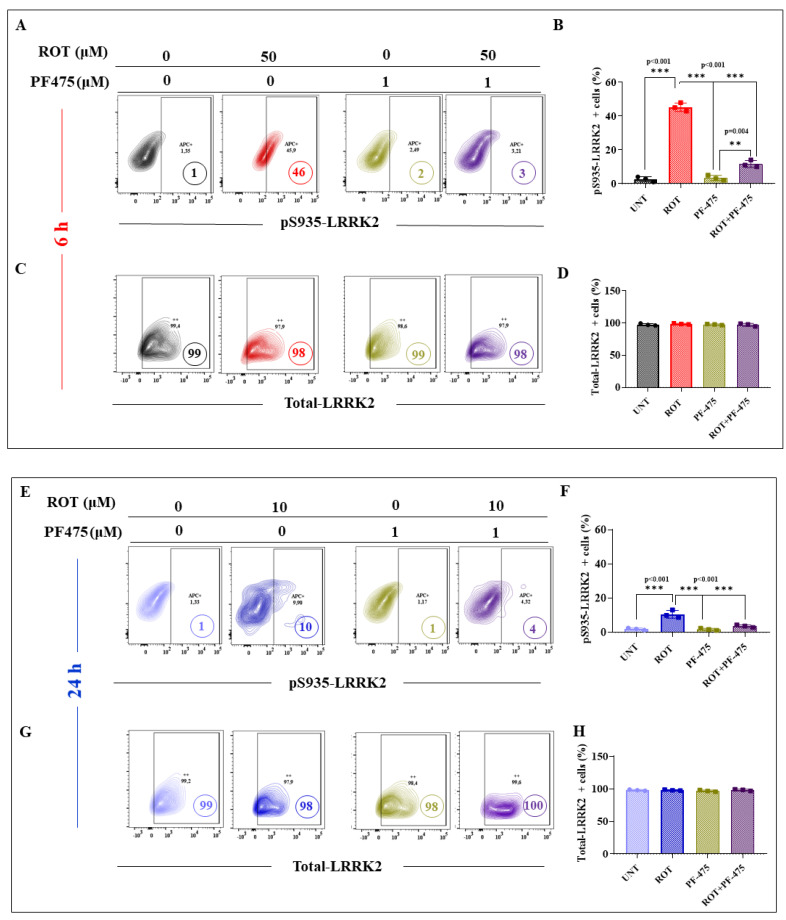
The inhibitor LRRK2 PF-06447475 (PF-475) inhibits pS^935^-LRKK2, and cleaved caspase 3 in DALNs treated with ROT. (**A**) Representative flow cytometry contour plots showing untreated DALNs or those treated with ROT (50 µM) in the absence or presence of PF-475 for 6 h, and stained with primary antibody against p-S^935^-LRKK2. (**B**) Percentage of p-S^935^-LRKK2-positive cells in untreated cells and cells treated with ROT and/or PF-475. (**C**) Representative flow cytometry contour plots showing untreated DALNs or those treated with ROT (10 µM) in the absence or presence of PF-475 for 6 h and stained with primary antibody against total-LRKK2. (**D**) Percentage of total LRKK2-positive cells in untreated cells and cells treated with ROT and/or PF475. (**E**) Representative flow cytometry contour plots showing untreated DALNs or those treated with ROT (10 µM) in the absence or presence of PF-475 for 24 h, and stained with primary antibody against p-S^935^-LRKK2; (**F**) Percentage of p-S^935^ LRKK2-positive cells in untreated cells and cells treated with ROT and/or PF-475. (**G**) Representative flow cytometry contour plots showing untreated DALNs or those treated with ROT (10 µM) in the absence or presence of PF-475 for 24 h and stained with primary antibody against total-LRKK2. (**H**) Percentage of total LRKK2-positive cells in untreated cells and cells treated with ROT and/or PF-475. (**I**) Representative flow cytometry contour plots showing untreated DALNs or those treated with ROT (50 µM) in the absence or presence of PF-475 for 6 h and stained with primary antibody against CC3; (**J**) Percentage of CC3-positive cells in untreated cells and cells treated with ROT and/or PF-475. (**K**) Representative flow cytometry contour plots showing untreated DALNs or those treated with ROT (10 µM) in the absence or presence of PF-475 for 24 h, and stained with primary antibody against CC3. (**L**) Percentage of CC3-positive cells in untreated cells and cells treated with ROT and/or PF-475. The data are presented as the mean ± SD of three independent experiments (*n* = 3). One-way ANOVA followed by Tukey’s test. Statistically significant differences: ** *p* < 0.01; *** *p* < 0.001. ++ symbol and numbers in the figure represent positive cellular population for the tested marker.

**Figure 11 ijms-24-15744-f011:**
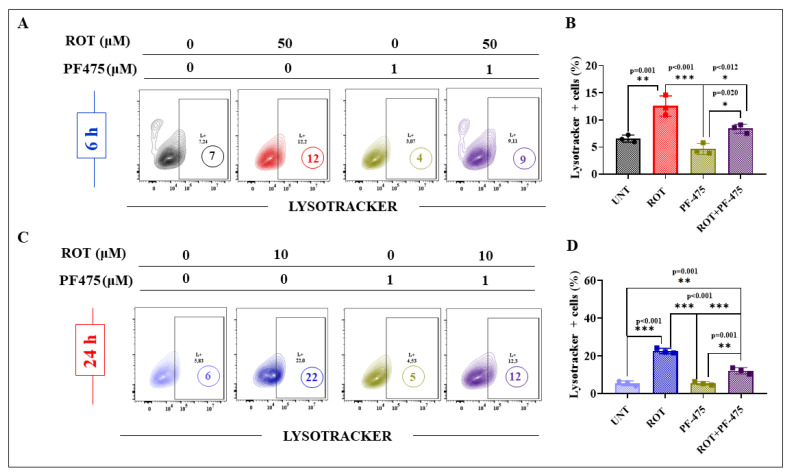
The inhibitor LRRK2 PF-06447475 (PF-475) affects the accumulation of lysosomes in DALNs treated with ROT. (**A**) Representative flow cytometry contour plots showing untreated DALNs or those treated with ROT (50 µM) in the absence or presence of PF-475 for 6 h and stained with Lysotracker^®^. (**B**) Percentage of Lysotracker^®^ stain-positive cells in untreated cells and cells treated with ROT and/or PF-475. (**C**) Representative flow cytometry contour plots showing untreated DALNs or those treated with ROT (10 µM) in the absence or presence of PF-475 for 24 h, and stained with Lysotracker^®^. (**D**) Percentage of Lysotracker^®^ stain-positive cells in untreated cells and cells treated with ROT and/or PF-475. The data are presented as the mean ± SD of three independent experiments (*n* = 3). One-way ANOVA followed by Tukey’s test. Statistically significant differences: * *p* < 0.05; ** *p* < 0.01; *** *p* < 0.001. L+ and numbers in the figure represent positive cellular population for the tested marker.

**Figure 12 ijms-24-15744-f012:**
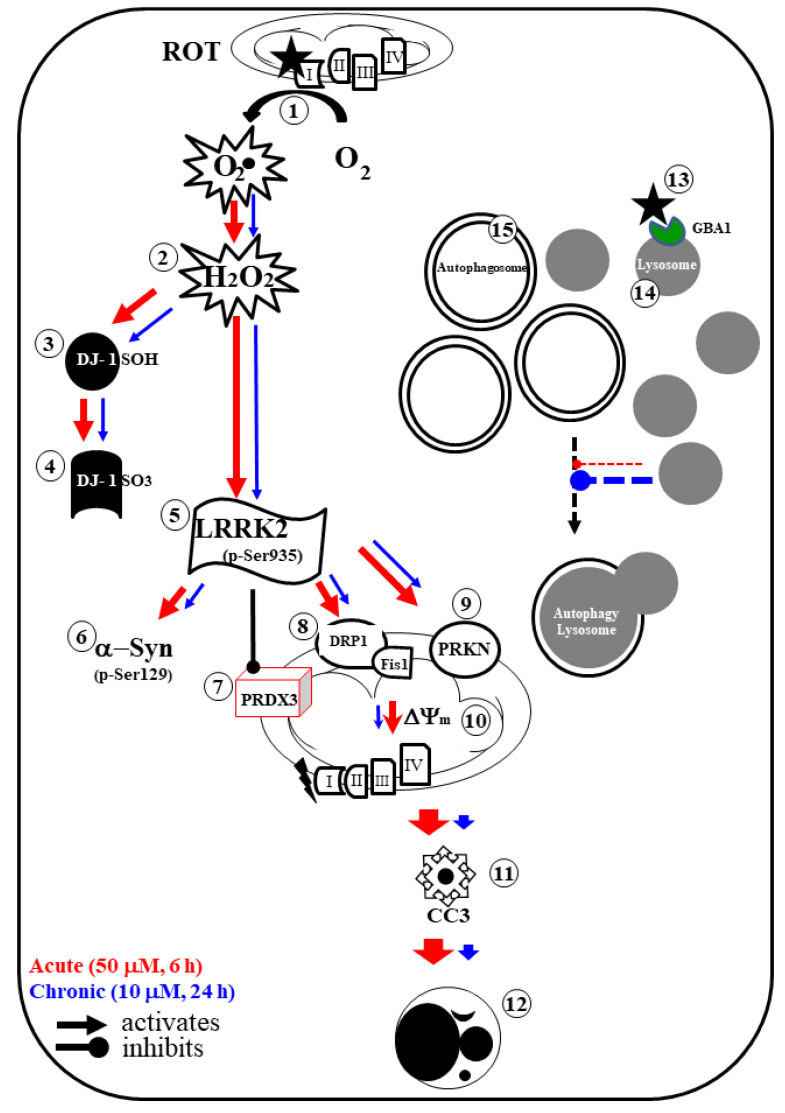
Schematic model of cell signaling induced by acute and chronic rotenone exposure in DALNs. Rotenone (ROT, black full star) binds to the ubiquinone binding site of mitochondrial complex I (NADH: ubiquinone oxidoreductase), thus preventing electron transfer via flavin mononucleotide (FMN) to coenzyme Q10 (**1**). Consequently, interruption of the electron transport chain concomitantly generates anion superoxide (O_2_^−^) and hydrogen peroxide (H_2_O_2_, **2**). This last compound is capable of oxidizing the stress sensor protein DJ-1Cys^106^-SH (**3**) into DJ-1Cys^106^-SO_3_ (**4**) and directly activates LRRK2 (leucine-rich repeat kinase 2) kinase by autophosphorylation (**5**). Once LRRK2 is phosphorylated at Ser^935^, the active p-(S-^935^)-LRRK2 kinase phosphorylates three major targets: (i) alpha-synuclein (α-Syn) at residue Ser^129^ (**6**), (ii) it inactivates protein peroxiredoxin 3 (PRDX3, **7**), thus preventing H_2_O_2_ catalysis [83]; (iii) p-(S-^935^)-LRRK2 activates the mitochondrial fission protein DLP-1 (dynamin-like protein 1, **8**), which, together with the fission protein-1 (Fis-1) receptor, induces an increase in the expression of parkin (PRKN, **9**), loss of mitochondrial potential (ΔΨm), fragmentation, and aggregation (**10**). Subsequently, the release of apoptogenic proteins (e.g., cytochrome C) results in the production of cleaved caspase 3 (**11**), which is responsible for chromatin condensation and DNA fragmentation (**12**), characteristics typical of apoptosis in DALNs. Alternatively, as shown previously, ROT binds and reduced the enzyme glucosylceramidase beta 1 (GBA1) (**13**). Moreover, the reduced catalytic activity of GCase results in a limited fusion of autophagosomes (**14**) and lysosomes (**15**), leading to their respective accumulation. Acute conditions: ROT 50 μM, 6 h (red arrow); chronic conditions: ROT 10 μM, 24 h (blue arrow). Arrow: activates; round pointed arrow: inhibits.

**Table 1 ijms-24-15744-t001:** Summary of the effect of acute and chronic ROT exposure on DALNs.

Condition/Variable	Acute (50 μM, 6 h)	Chronic (10 μM, 24 h)	Acute vs. Chronic
ΔΨ_m_	−49%	−13%	−39%
PRKN	+514%	+363%	+16%
DRP1	+85%	+50%	+41%
DJ-1Cys^106^-SO_3_	+1300%	+583%	+71%
pSer^935^-LRRK2	+750%	+266%	+210%
pSer^129^-α-Syn	+550%	+183%	+53%
CC3	+1500%	+500%	+78%
LC3B	+460%	+400%	−30%
Lysosome	+104%	+146%	−18%

## Data Availability

All relevant data are within the manuscript.

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
