# Peer review of "High Yield of Functional Dopamine-like Neurons Obtained in NeuroForsk 2.0 Medium to Study Acute and Chronic Rotenone Effects on Oxidative Stress, Autophagy, and Apoptosis"

_ijms, 2023, doi:10.3390/ijms242115744_

Round 1

Reviewer 1 Report

Comments and Suggestions for Authors

In this manuscript, the authors show a new method for neuronal differentiation using MenSCs and they achieve a high yield of dopaminergic neurons in culture. In addition, they test the effect of two concentrations and times of rotenone and recapitulate the effect on mitochondrial defects, apoptosis and autophagy levels. The work described here is interesting and it can be used for future reference in follow up studies. However, I find that there is not enough evidence in this manuscript to conclude that ROT induces apoptosis independently of the autophagy effects. Further characterization of autophagy is strongly recommended before publication. In addition, I also suggest an improvement in the manuscript layout before publication. Apart from this, a few points should be taken into consideration.

-          Abstract is too long. Remove data numbers from abstract and introduction. In abstract, summarise findings. Avoid sentences longer than 3 lines (e.g. line 28-40). In addition, sentence:

-          Abstract: “Our data suggest that apoptotic cell death prevails in both acute and chronic ROT conditions independently of autophagy dysregulation”: Rephrase, this is not shown here. Flux or LC3II/I ratio was not assessed or autophagy flux. Also, to properly address the dependency on autophagy and apoptosis via ROT: knockdown ATG7 or any autophagy protein and analyse if ROT still induces apoptosis/mitochondrial defects.

-          Abstract figure does not have a legend.

-          Line 87- missing “)”

-          Reference 27- rotenone was not used as a stressor in this paper.

-          ROS and OS abbreviations have not been defined.

-          Rephrase sentence Line 101-103.

-          Rephrase sentence “Thefore, it is not yet determined whether autophagy induces, reduces, or has a protective effect on DAergic neuron cell death”: evidence shown in the introduction does not correlate with this. Indeed, the neuroprotective effect of autophagy is fairly established in the field: https://doi.org/10.1038/s41598-018-21325-w;  https://doi.org/10.1083/jcb.201910133; https://doi.org/10.1038/cddis.2014.358

-          Rephrase sentence “Thefore, we hypothesized that acute or chronic ROT exposure induces apoptosis independently of the accumulation of autophagosomes and lysosomes in DALNs”.

-          Last paragraph in the introduction is too long, sentences are too long. Rephrase.

-          LC3 II accumulation is not a marker of dysfunctional autophagy, and in addition, you are using a LC3B antibody, so both LC3I and LC3II will be recognised.

-          Figures have too many letters, this is particularly confusing in Figure 3 and 4. Every images does not need a letter as colours and headings are properly used. They are also quite big (Figure 1 can be splitted in two figures).

-          Results 2.1. Are the cells expressing midbrain markers?

-          Results 2.1. Figure 1F-I. Cells do not look very neuronal, have you stained for TUJ1 or MAP2? Any phase contrast images?

-          Results 2.2. Line 205, remove extra space.

-          Results 2.2. Concentration and time curse is needed here. Why were these two conditions the best for acute and chronic? Was ROT tested for more than 24h?

-          Figure 2. Do the graphs represent 1 out of 3 independent experiments? (information obtained from the legend) If so, normalise the three experiments and show that data.

-          Results 2.2. Statement: “But also a significant statistically difference between DALNs exposed to ROT …”, this was not analysed in the graphs or discussed in the legend: for that do Anova test comparing all conditions in the same graph.

-          Figure 3. Figures F,L, S and Y are too small.

-          Results 2.3. What are the total LRRK2 and alpha synuclein levels? The increase in phosphorylated protein levels could be explained by an increase in total levels.

-          Line 305, remove alpha-Syn at the end of the sentence.

-          Figure 4, too big, split in 2.

-          Figure 5. Rename title, there is an increase in lipidated LC3, expression is not addressed here.

-          Results 2.5 and Figure 6. Flux is not analysed in this manuscript. Rename figure title. To address flux, either you would need to look at the ratio of LC3I/LC3II and also in the presence or absence or a lysosomal inhibitor such as BafA1. Comparing LC3B levels and lysosomal levels to rapamycin or BafA1 does not indicate if it is inducing or inhibiting flux. I would recommend using an autophagy or mitophagy flux reporter such as mRFP-GPF-LC3B or mitoQC or looking at LC3II/I +/- BafA1 by immunoblotting: doi: 10.3390/ijms18091865.

-          Discussion- Line 476. Rephrase, reasons as detailed above. I agree that it is probably the right mechanism based on published data, but the data from this paper does not contribute to this.

-          Figure 7. Rotenone inhibiting autophagy: this has not been shown in this manuscript. Only the accumulation of LC3II and lysosomes, which is also observed with autophagy inducers.

-          Conclusion. As described above, evidence here does not suggest that targeting apoptosis rather than autophagy is a potential treatment strategy for PD.  

Reviewer 2 Report

Comments and Suggestions for Authors

This arduous experimental study carried out by Quintero-Espinosa and collaborators meets the requirements to be published. Not only are they perfecting a culture medium to obtain a good percentage of dopaminergic cells, but they are also developing a new model for the development of possible therapies in this context that can be brought to the clinic.

Round 2

Reviewer 1 Report

Comments and Suggestions for Authors

I am pleased to say that all my comments have been addressed, the manuscript now looks very completed.